# Cortisol and the Dexamethasone Suppression Test as a Biomarker for Melancholic Depression: A Narrative Review

**DOI:** 10.3390/jpm13050837

**Published:** 2023-05-16

**Authors:** Martin M. Schumacher, Jacopo Santambrogio

**Affiliations:** 1Independent Researcher, Linsenackerweg 4, 4450 Sissach, Switzerland; martin.schumacher@sunrise.ch; 2Adele Bonolis AS.FRA. Onlus Foundation, 20854 Vedano al Lambro, Italy; 3Presidio Corberi, ASST Brianza, 20812 Limbiate, Italy

**Keywords:** dexamethasone suppression test, Cushing’s syndrome, HPA-axis, depression, melancholia, cortisol, suicide, drug development

## Abstract

The dexamethasone suppression test (DST) assesses the functionality of the HPA axis and can be regarded as the first potential biomarker in psychiatry. In 1981, a group of researchers at the University of Michigan published a groundbreaking paper regarding its use for diagnosing melancholic depression, reporting a diagnostic sensitivity of 67% and a specificity of 95%. While this study generated much enthusiasm and high expectations in the field of biological psychiatry, subsequent studies produced equivocal results, leading to the test being rejected by the American Psychiatric Association. The scientific reasons leading to the rise and fall of the DST are assessed in this review, suggestions are provided as to how the original test can be improved, and its potential applications in clinical psychiatry are discussed. An improved, standardized, and validated version of the DST would be a biologically meaningful and useful biomarker in psychiatry, providing a tool for clinicians caring for depressed patients in the areas of diagnosis, treatment, and prognosis, and predicting the risk of suicide. Additionally, such a test could be a crucial part in the generation of biologically homogenous patient cohorts, necessary for the successful development of new psychotropic medications.

## 1. Introduction

With the advent of modern psychotropic medications, starting in the early 1950s, psychiatry became more biology-oriented, with high expectations for identifying the somatic etiologies of the endogenous psychoses and corresponding biomarkers. The neurotransmitter-based hypotheses of the etiologies/pathophysiologies of depression and schizophrenia further fueled this development. Of the nervous, immune, and endocrine systems, the nervous system has received much attention and importance in biopsychiatry, but no robust biomarkers associated with neurotransmitters have yet been found. More recently, the important role of the immune system in mental disorders has been acknowledged, with autoimmune encephalitides being identified as causes of different neurological and psychiatric disorders in some patients. This is currently a very active field and many autoantibodies can be considered as promising biomarkers.

Disorders of the endocrine system, however, were the first to be recognized as strongly related to mental disorders. Psychiatrists Maxime Laignel-Lavastine (1875–1953), Manfred Bleuler (1903–1994), and Edward Sachar (1933–1984), and neurosurgeon Harvey Cushing (1869–1939) were among the early leaders in the area where endocrinology and psychiatry intersect. Although all endocrine systems have an impact on mental wellbeing, the hypothalamic–pituitary–adrenal (HPA) axis, in particular, has received much attention in psychiatry [1]. Early research in this field started in the 1960s, headed by Edward Sachar and Bernard Carroll (1940–2018). The dexamethasone suppression test (DST) was introduced in endocrinology for the diagnosis of Cushing’s syndrome in 1960 [2]. Its importance for psychiatry was recognized by Carroll; in 1968, he started studying cortisol in plasma, urine, and cerebrospinal fluid; varying the times and dosages of glucocorticoid administration; examining different sampling schedules; optimizing the statistical handling of these complex measures; and exploring the differences between psychiatric patients with melancholic depression, patients with other psychiatric disorders, and normal subjects. It is important to distinguish melancholic depression (melancholia, endogenous depression) from exogenous forms of depression (i.e., neurotic, reactive). Melancholia is considered to be a biological disease characterized by a recurrent course, familial aggregation, and a pronounced component of psychomotor disturbances (i.e., retardation and/or agitation of mental and physical activities). Carroll observed that in the majority of patients with melancholic depression, the application of dexamethasone, a synthetic high-potency glucocorticoid with a long biological half-life, approximately 25 times more active than endogenous cortisol, did not suppress the secretion of 11-hydroxycorticosteroids (e.g., cortisol) [3]. He led further research that, in 1981, culminated in the seminal article, A Specific Laboratory Test for the Diagnosis of Melancholia, published in the Archives of General Psychiatry [4], in which the authors proposed the DST for a laboratory-based diagnosis of melancholia, so the DST can be considered as the first biomarker in psychiatry. This article was met with great interest and enthusiasm by the psychiatric community and triggered many studies by other researchers using the DST as a diagnostic tool in psychiatry. Unfortunately, the results of these studies were inconsistent and, for the most part, did not confirm Carroll’s findings and claims. The reasons for this are various and include the use of divergent versions of the DST (according to Helena Kraemer there were “*almost as many DSTs as there are DST studies*” [5] and significantly different patient populations. In 1980, the American Psychiatric Association (APA) published DSM-III, introducing the very broad major depressive disorder (MDD) construct. As many researchers subsequently applied the DSM-III criteria for patient selection, the DST lost much of its sensitivity. As explained by Shorter and Fink [1] and Lutz [6], there were various factors that contributed to DST’s ultimate fall from grace including personal agendas, psychiatric community dynamics, political factors, the prevailing *zeitgeist*, and the unfavorable judgement of the APA. An improved, standardized, and validated version of the DST would be a useful tool in clinical psychiatry and also for drug development.

With regard to biomarkers, there are different types available including diagnostic, prognostic (for the natural evolution of the condition), mechanistic (related to the pathophysiology), and predictive (regarding the response to treatment). To date, there are no universally accepted biomarkers in psychiatry with satisfactory performance statistics. Kraemer [7] provides an authoritative introduction to biomarkers in psychiatry.

## 2. Hypercortisolism, the HPA Axis, Cushing’s Syndrome, and Depression

Hypercortisolism is common in patients with severe depression [8,9,10,11]; it manifests in elevated levels of cortisol in the serum, cerebrospinal fluid (CSF), saliva and/or 24 h urine and elevated levels of corticotropin-releasing hormone (CRH) in the CSF.

Research in this field started in the 1960s [3,12] and a comprehensive review was published recently [13]. On the other hand, depression and, to a lesser extent, mania and anxiety are also very common in patients suffering from Cushing’s syndrome (CS) [14,15]. In most CS patients, hypercortisolism is due either to hypersecretion of the adrenocorticotropic hormone (ACTH) by a pituitary tumor, to ectopic ACTH secretion from an extrapituitary neoplastic lesion, or to autonomous cortisol secretion by an adrenal tumor.

A detailed list of CS etiologies, which fall into two major categories, endogenous and exogenous, can be found in Kannan [16]. Distinguishing between endogenous and exogenous CS can be challenging for both the endocrinologist and psychiatrist, with the risk of a wrong diagnosis having negative consequences for the patient. This situation becomes even more complicated as some patients with physiological hypercortisolism exhibit only minimal physical CS features.

In certain subclinical forms of endogenous CS [17,18], psychiatric symptoms are their first and only manifestation [19]. A recurrent form of CS, cyclical Cushing’s syndrome, which can closely mimic recurrent mood disorders such as melancholia, has been reported for all CS etiologies. Irrespective of age, primary pigmented nodular adrenocortical disease (PPNAD) and isolated micronodular adrenocortical disease are also often cyclic [19]. These are important observations, as melancholic depression (according to the definition of Kraepelin et al. [20]) is also a periodic disease.

Exogenous CS patients fall into two groups, iatrogenic and factitious. The former is due to prolonged treatment with corticosteroids or ACTH, while the latter, usually called pseudo-Cushing’s syndrome (i.e., physiologic or non-neoplastic CS), is mainly caused by neuropsychiatric disorders, polycystic ovary syndrome (PCOS), obesity, poorly controlled diabetes mellitus, alcoholism, or eating disorders [21].

All CS etiologies have an excessive secretion of cortisol as a common final pathway. The clinical pictures with regard to the physical and mental signs and symptoms are due to the underlying hypercortisolism. However, these do not allow for the identification of their specific etiology and call for further clinical investigations (e.g., imaging, biopsies, and hormone analyses).

The treatment of CS should target the underlying etiology. Iatrogenic CS and the mental symptoms caused by prolonged treatment with glucocorticoids [22] cease when the drug is withdrawn. Such treatment, which targets the reduction in glucocorticoid synthesis or action, either with metyrapone, ketoconazole, or mifepristone, as opposed to treatment with antidepressant drugs, is generally successful in relieving depressive symptoms as well as other disabling symptoms [23,24]. Following successful surgical treatment of hypercortisolism, both physical and psychiatric signs and symptoms substantially improve [25]. These findings suggest that hypercortisolism might be the cause of the observed psychopathology.

In healthy subjects, the release of CRH and ACTH is regulated by cortisol via a negative feedback mechanism, while in patients with Cushing’s disease (CD) and other subtypes of CS, this feedback is impaired, leading to the secretion of excess cortisol. In healthy subjects, dexamethasone (DEX) acts on the hypothalamus and pituitary, suppressing the secretion of cortisol, but this does not happen in patients with CS. Interestingly, nonsuppression was also observed in some patients with exogenous CS (pseudo-Cushing’s syndrome) subjected to the DST.

The psychiatric symptoms and their frequency observed in patients with endogenous CS and those with major depression were described by Murphy [23]. The prevalence of typical symptoms such as depressed mood, decreased energy, irritability, insomnia, and impaired memory are increased to a similar extent in both diseases.

Many organic illnesses have a recurrent or intermittent course. Organic illnesses can also cause recurrent mental symptoms, suggesting a “psychogenic” disorder and making the detection of the underlying disease more difficult. Gustave Newman lists the following related diseases: multiple sclerosis, acute intermittent porphyria, pheochromocytoma, systemic lupus erythematosus, pancreatitis, herpes simplex encephalitis and episodic dyscontrol syndrome [26]. Other autoimmune diseases, intermittent Cushing’s syndrome, and other diseases can be added to this list. This observation supports the possibility that even mental disorders, such as melancholia and bipolar disorder, which by definition have a recurrent course, can be mimicked by somatic diseases.

A combined DEX/corticotropin-releasing hormone (CRT) test [27], the cortisol-awakening response (CAR) [28], and individual cortisol levels or profiles [29,30] were also proposed for diagnosing depression in addition to Carroll’s DST but are not discussed in this review.

## 3. The DST and Psychiatric Nosology

Carroll, in his 1981 publication [4], proposed the DST as a tool for the diagnosis of melancholia. Obviously, the definition of melancholia and the reliability of this diagnosis are of utmost importance for the performance of the DST (i.e., its sensitivity and specificity).

What is melancholia? This question might be considered naïve, but it is nevertheless very relevant because there is still no consensus in the psychiatric community regarding melancholia as a mental disease *sui generis* or the characteristics uniquely describing it. Melancholia (also called endogenous, endogenomorphic, or vital depression) has been described as a clinical entity for millennia and was widely accepted by the physicians working in lunatic asylums of the past. Emil Kraepelin considered melancholia to be part of manic-depressive insanity (MDI) [20,31]. This definition of melancholia includes psychotic depression, bipolar depression, mixed bipolar depression, and probably also schizoaffective psychosis. In contrast to ordinary depression, which is mainly phenomenologically characterized by different mental symptoms, melancholia has a strong somatic component comprising a recurrent course, familial aggregation, and pronounced psychomotor disturbances. A melancholic episode often occurs without any external (psychosocial) triggers and is inert to psychotherapeutic interventions.

With the advent of DSM-III in 1980, melancholia was grouped together with other neurotic/reactive depressions under the label of major depressive disorder (MDD). This resulted in a very heterogeneous entity that continues to hinder progress in depression research and psychopharmacology and has been lamented by many experts [32,33,34]. Indeed, several experts have called for melancholia to be reinstated as a valid disease entity [35,36]. It is also described comprehensively in the literature [37,38].

Currently, the most reliable diagnostic tool for melancholia is probably the CORE measure [36,37] of psychomotor disturbance, which is based on 18 signs (not symptoms) assessed by an experienced clinician. These signs belong to the subscales of agitation (facial agitation, motor agitation, facial apprehension, stereotyped movement, and verbal stereotypy), retardation (slowed speed of movement, slowing of speech, delay in motor activity, bodily immobility, delay in verbal responses, facial immobility, and postural slumping) and noninteractiveness (nonreactivity, inattentiveness, poverty of associations, shortened verbal responses, and impaired spontaneity of speech). A value is attributed to each sign (zero if absent; one if present). These values are then added to obtain the total CORE score (range: 0 to 18). A score of eight or more is needed for the diagnosis of melancholia. There is also a more refined CORE measure, graded according to the severity of the signs (absent: 0, present: 1 to 3) available (range: 0 to 54). The reliability and validity of the CORE measure has been assessed in many international studies [37,39].

Carroll et al., investigating the performance of the DST, made a clinical diagnosis of melancholia as described in the article Diagnosis of Endogenous Depression published in the Journal of Affective Disorders [40]. In addition to the application of a structured psychiatric interview (Schedule for Affective Disorders and Schizophrenia (SADS)), the patient’s previous psychiatric history, family history, and past hospital records were taken into consideration. The major diagnostic features of endogenous depression (i.e., melancholia) were listed as: (i) history of mania, hypomania, or endogenous depression; (ii) definite family history; (iii) severe agitation or retardation; (iv) depressive psychosis; (v) pervasive anhedonia; (vi) definite pathological guilt. The severity of depression was quantified with the Hamilton rating scale and the Carroll self-rating scale [40].

Other researchers have mostly used only symptom-based tools such as the Research Diagnostic Criteria (RDC) or the DSM-III, though some have preferred the Newcastle scale [41,42,43,44]. It must be clearly stated that the vast majority of other researchers investigating the DST have not used the Carroll–Feinberg definition and diagnostic criteria of melancholia, applying broader (and less appropriate) definitions. The use of these different tools partially explains the inconsistent results across studies.

Mark Zimmerman et al. from the University of Iowa examined the relationship between the performance of the DST and four definitions of endogenous depression, namely DSM-III, Feinberg and Carroll, Newcastle, and RDC [44]. They found rather similar percentages (36–48%) of nonsuppressors in groups of patients diagnosed with “definite endogenous depression”.

Unfortunately, there is no gold standard for the tools used for the diagnosis of melancholia; Hui and Zhou of the Division of Biostatistics at the Indiana University School of Medicine reviewed the statistical methods developed to estimate the sensitivity and specificity of screening or diagnostic tests when the fallible tests are not evaluated against a gold standard [45].

## 4. The Original DST as Used by Carroll et al. [4]

Up to March 2023, Carroll et al.’s seminal article [4] has been cited 2497 times. The study involved a total of 368 patients (180 inpatients and 188 outpatients; 215 patients with a diagnosis of melancholia and 153 patients with a diagnosis of nonmelancholic depression or other mental disorders) and a control group of 70 normal subjects. The parameters and limitations of the test, and the authors’ findings and claims, are listed below.

### 4.1. Test Parameters

The dose and timing of dexamethasone (DEX) were 1 mg p.o. at 11 p.m.; two blood samples were taken at 4 p.m. and 11 p.m. after the administration of DEX for the quantification of cortisol; nonsuppression was defined by a cortisol concentration > 5 μg/dL at either of the two time points. The plasma cortisol was quantified by the competitive protein binding method. Patients with specific somatic diseases were not subjected to the test. The diagnosis of melancholia was made through a composite clinical assessment [40].

### 4.2. Test Results

The DST identified melancholic inpatients with a sensitivity of 67% and a specificity of 96% (for melancholic outpatients, a sensitivity of 49% was obtained). The outcome of the DST was not related to age, sex, or the recent use of psychotropic medication. In patients with a psychiatric diagnosis other than melancholia and in normal subjects, the specificity of the DST was 96% in both cases. Nocturnal (11.30 p.m.) pre-DEX plasma cortisol levels had less diagnostic power (i.e., lower sensitivity and specificity) than the DST. The group of melancholic patients was heterogeneous with respect to neuroendocrine function as assessed by the DST. A negative DST result did not rule out the diagnosis of melancholia. Abnormal DST responses (i.e., nonsuppression) returned to normal upon recovery from the condition.

One year later, Carroll published a comprehensive review of the DST in the British Journal of Psychiatry [46], summarizing the results obtained by different groups and comparing them to his own findings. Different clinical uses of the DST, sources of variation in sensitivity (DEX dose, post-DEX schedule/timing of blood sampling, post-DEX plasma cortisol threshold, and diagnostic criteria) and applications to nosology were discussed (see also [47]).

### 4.3. Main Takeaways

The DST is a specific episode-related biological marker of melancholia (i.e., it is a state-dependent biomarker, not a trait marker of melancholia *per se*). A diagnostic trait biomarker identifies the disease at all times; a state biomarker only identifies the disease when the disease is active. This distinction is very important in the case of episodic diseases such as melancholia. An additional benefit of the DST as a state biomarker is its usefulness for monitoring the success of treatment (i.e., normal suppression in remission). The clinical uses include the assessment of treatment response to electroconvulsive therapy (ECT) or pharmacotherapy with tricyclic antidepressants (TCAs), the prediction of relapse, and as an indicator of suicide risk. The diagnostic confidence of the DST depends on the prevalence of cases (i.e., patients with melancholia). Therefore, the DST is not suitable in situations with low prevalence, i.e., for screening purposes in a general outpatient setting [48].

In 1985, Carroll published another important paper, Dexamethasone Suppression Test: A Review of Contemporary Confusion, in which he defended the DST, writing in the summary, “Reasons for the current controversy and confusion about the dexamethasone suppression test (DST) are reviewed, and basic axioms regarding use and interpretation of the test are reiterated. Problems with reliability and validity of current diagnostic systems limit their use as ‘gold standards’ for evaluation the DST; accurate evaluation must await follow-up and treatment response studies. Interpretation of DST results in specific patients requires common sense, consideration of the clinical context, and attention to technical factors. While its ultimate significance is not yet known, the DST, like other laboratory tests, may help to resolve uncertainty in clinical diagnosis. Perhaps most important, it may help to refine current paradigms for psychiatric nosology and diagnosis.” [49]. This 12-page article in the *Journal of Clinical Psychiatry* contains many sound arguments concerning the use and interpretation of the DST as well as pertinent comments on the contradictory findings of other research groups.

## 5. The DST as Applied by Other Researchers


*“How does one validate a biological marker of endogenous depression when a valid clinical definition does not exist?”*
Mark Zimmerman [42]


*“What is both impressive and dismaying in reviewing the published DST evaluations is that there is not one DST, but almost as many DSTs as there are DST studies. Mixing together different tests is a major source of the confusion in the evaluation of the DST, particularly in the reviews of the test.”*
Helena Kraemer [5]


*“A modified dexamethasone suppression test (DST) has had unprecedented evaluation among biologic tests proposed for clinical use in psychiatry. It has not proved to reflect pathophysiologic changes at the level of the central nervous system or pituitary, and tissue availability of dexamethasone itself may contribute to test outcome.”*
George Arana [50]

The DST has been used by many research groups, and a plethora of related studies with widely differing results have been published in peer-reviewed journals. Although most have reported a significantly elevated proportion of nonsuppression in depressed patients with melancholic features, the sensitivities are usually lower than the 67% reported by Carroll et al. and cover a broad range. Additionally, the proportions of nonsuppressors in patient populations with psychiatric diagnoses apart from melancholia and even normal controls is often substantially larger than the 4% they reported (i.e., specificity = 96%) (see also [48,50,51,52,53,54,55,56]).

Table 1 shows the DST results of several research groups based on the Research Diagnostic Criteria (RDC) for the definition of major depressive disorder—endogenous type (MDD-ET). Different test parameters, as shown in the table, were used. The proportion of nonsuppressors (i.e., sensitivity) with MDD-ET covers the range of 22% to 81% (mean = 40%). For control subjects, the proportion of nonsuppressors (specificity = 100 − % nonsuppression) is 4–15% (mean = 10%) and for patients with other psychiatric illnesses 0–37% (mean = 18%). See Insel and Goodwin (1983) for the studies cited in the table [57].

Table 2 below shows the percentages of nonsuppression for a series of psychiatric disorders and multiple threshold values of the DST (adapted from Evans and Nemeroff [58]; a definition of the different diagnoses in the table can be found in this article). A pronounced association between the severity of different mood disorders and the percentage of nonsuppression can be observed.

Similar results have been obtained by other researchers, as shown in Table 3 (adapted from Murphy [23]).

Patients with mixed bipolar, psychotic depression, and melancholia show the highest rate of nonsuppressors [59]. This indicates that these conditions are not independent (categorical) diseases but are part of a continuum.

Important findings from various research groups are summarized below:Parker and Hadzi-Pavlovic’s CORE rating scale is probably the best tool currently available for the diagnosis of melancholia. A group of 100 mildly to severely depressed inpatients were assessed with the CORE measure and subjected to the DST. An almost perfect linear relationship between the CORE scores and the percentage of nonsuppression was found (30% at a CORE of 2 to 90% at a CORE of 32). In the same patient population, a similar, although weaker, correlation was found when Newcastle scores were used [37].Dwight Evans from the University of North Carolina at Chapel Hill applied the 1 mg overnight DST test to 166 depressed (according to DSM-III criteria) inpatients [60]. Using the 5 µg/dL threshold for the definition of DST nonsuppression, he found a marked dependence of the proportions of nonsuppressors on the type of depression: “depressive symptoms” (14%), MDD without melancholia (48%), MDD with melancholia (78%), and MDD with psychosis (95%). Interestingly, he also reported a high rate of 17% of subclinical autoimmune thyroiditis in the nonsuppressors of the same patient group (vs. 3% in the suppressor group).In 1987, Helmfried Klein, a member of Hanns Hippius’ research group in Munich, published a monograph on biological markers in affective disorders in which he comprehensively reviewed the published literature regarding the sensitivity and the specificity of the DST in different patient populations and added his own research results [61]. His findings (averages) were based on: (i) all studies (irrespective of the parameters of the DST used), and (ii) only those studies with DST parameters as proposed by Carroll et al. (i.e., 1 mg DEX p.o., serum cortisol determination at 4 p.m. post-DEX; threshold > 5 µg/dL cortisol).A selection of his results is provided below.
-SpecificityHealthy controls:In 15 studies (# patients N = 646), a mean specificity of 93.6% was found. In a subset of seven studies (# patients N = 305) applying the DST parameters used by Carroll, the average specificity was 92.1%.Psychiatric patients (diagnosis other than depression):In 20 studies (# patients N = 656), a mean specificity of 76% was found (i.e., 24% nonsuppression). In a subset of 10 studies (# patients N = 292) applying the DST parameters used by Carroll, the specificity dropped to 69%.-SensitivityA total of 10 studies were conducted to compare sensitivity in 996 patients, of whom 608 were suffering from endogenous depression and 388 from nonendogenous depression. In the endogenous depression group, the sensitivity was 43%, while in the nonendogenous depression group, it was 24%.A total of 18 studies were conducted with al 1219 depressed patients in 2 heterogeneous diagnostic groups. Patients with primary, endogenous, and psychotic depression were assigned to Group 1; while patients with secondary, nonpsychotic, nonendogenous, minor, bipolar, and neurotic/reactive depression were assigned to Group 2. The proportion of nonsuppressors in Group 1 was 56% and that in Group 2 was 23%. The overall proportion of nonsuppressors in both groups was 40%. In a subset of seven studies (N = 408) using Carroll’s DST parameters, 72% and 42% of nonsuppressors were found for Groups 1 and 2, respectively. Obviously, the diagnoses represented in the groups were very heterogeneous, and the diagnostic criteria were highly variable.-Other relevant resultsContrary to Carroll’s findings, in his patient cohort, Klein observed a statistically significant difference of the post-DEX dexamethasone concentrations at 4 p.m. between men and women, which was also present in the corresponding cortisol concentrations [61]. The results for a dose of 1 mg DEX are (i) in females 62.7 ng/dL DEX and 2.25 µg/dL cortisol, and (ii) in males 115.9 ng/dL DEX and 0.89 µg/dL cortisol.The proportion of nonresponders correlated with the severity of depression, as measured by the Hamilton Depression Scale (HAMD).In 1987, the World Health Organization published a collaborative study of the DST involving 12 countries and 13 research centers [62], in which the DST was applied to 543 patients suffering from major depressive disorders and 246 healthy controls. The mean post-DEX cortisol concentrations of the depressed patients (range: 2.3–11.6 µg/dL, mean: 7.1 µg/dL) and the healthy controls (range: 1.3–4.9 µg/dL, mean: 2.5 µg/dL) showed vast variability between the centers, indicating a lack of standardization of the DST and the patient cohorts. This was also reflected in the observed mean percentages of nonsuppressors for depressed patients (range: 15–71%, mean: 46%) and healthy controls (range: 0–40%, mean: 9%). The mean post-DEX cortisol concentrations and the mean percentages of nonsuppression were strongly correlated. Although there was substantial variation between the different centers, the general observation of hypercortisolism and a higher number of positive DSTs in depressed patients was confirmed.A large cohort study conducted in the Netherlands showed how the DST can be misapplied in the context of depression [63]. A total of 1588 patients (308 controls and 1280 patients with MDD) were subjected to the DST, and more than 11,000 cortisol determinations (in samples drawn at 7 different time points) were performed. None of Carroll et al.’s 1981 test parameters, patient population characteristics, or the depression definition were respected [4]. The control group showed a higher proportion of nonsuppressors (14.9%) than the patients with current MDD (11.0%) and those with remitted MDD (13.8%). The results were judged to be inconclusive by the authors.

## 6. DST Features and Limitations

Why was the DST rejected by the psychiatric community more than 30 years ago? As Shorter and Fink [1] and Lutz [6] have pointed out, there were a number of reasons, most of which were unrelated to the DST as a scientific procedure. These authors attributed the rejection of the DST to a combination of various factors, including the impact of the *zeitgeist*, personality conflicts, and negative dynamics in the psychiatric community. However, a significant issue was that Carroll’s results were not replicated by a majority of other researchers. The attempts of the World Health Organization [62] and the American Psychiatric Association (APA) [64] to settle this issue through dedicated task forces were not successful.

As Lutz correctly mentioned, the DST was never fully validated or standardized [6]. It must be remembered, however, that 40 years ago, the possibilities of a clinical chemistry laboratory were more limited than they are today.

As mentioned earlier, many factors unrelated to the DST, already recognized by Carroll [46,49] and later amended by other researchers, confounded the outcome. Liebl and Klein have also discussed several of these factors [61,65]. The most relevant areas of discord are discussed below, with recommendations for improvement.

### 6.1. Dosage and Time of Application of Dexamethasone (DEX)

The doses used for the DST by different research groups were mostly in the range of 0.5 to 2 mg DEX, given orally at 11 p.m. Carroll et al. determined the optimal dosage to be 1 mg; others opted for 2 mg. Hunt et al. sequentially applied DEX doses of 0.5 and 1.5 mg to depressed patients and controls, obtaining significantly different sensitivities of the DST [66]. Neither the optimal dose nor the best application time have been determined by a sound scientific procedure. 

Recommendation: as the dose and application time interact, they should be optimized together.

### 6.2. Quantification of Cortisol

Carroll used the competitive protein-binding method (CPB), while the majority of other researchers used the more specific radioimmunoassay (RIA), but with different antibodies. Ritchie et al. compared the CPB to 16 different commercial RIAs using post-DEX plasma pools [67]. For a fixed cortisol concentration of 5 µg/dL, the corresponding concentrations determined by the different RIAs were in the range of 4.3 to 8.7 µg/dL. This finding explains some of the variance in the sensitivity and specificity of the DST found in the different studies. 

Recommendation: a standardization of the analytical procedure and a lab-specific threshold in each laboratory should be established. Today, more powerful methods for the quantification of cortisol are available. The liquid chromatography tandem mass spectrometry (LC-MS/MS) method is both highly sensitive and specific [68]. An additional bonus is the capacity to quantify additional important analytes such as cortisone and dexamethasone in the same analytical run.

### 6.3. Factors Influencing the Cortisol Concentration

One of the most important confounding factors affecting the post-DST cortisol concentration is the DEX blood level at the time of measurement [69,70,71,72]. A standard dose of DEX results in a substantial inter-patient variation. This was also recognized by Carrol and colleagues [73,74]. However, to the best of our knowledge, no comprehensive study of the DST incorporating the DEX concentration was ever performed. A study from the laboratory of Robert Rubin found that post-DEX serum dexamethasone concentrations significantly influenced DST outcome only when they were below a certain threshold level [75]. If the DEX concentrations were too low, a substantial proportion of false-positive DSTs resulted. Low DEX levels have been attributed to a polymorphism of the metabolizing enzyme cytochrome CYP3A4 causing rapid metabolism or to the use of drugs that induce the production of this enzyme. Other potential causes include low gastrointestinal absorption and increased distribution of DEX due to low albumin binding. The elimination of DEX can also be affected by reduced liver and kidney functions [76]. False-negative DST results can also be caused by excessively high concentrations resulting from an impaired DEX metabolism and other factors. This significant interpatient variation in dexamethasone was one of the most important scientific reasons for the final rejection of the DST. 

Recommendation: as only free cortisol (and free dexamethasone) is biologically active, the concentration of corticosteroid binding globulin (CBG) and other binding proteins is also of importance and must be taken into consideration [77]. Concentrations of free cortisol (and dexamethasone) could be directly obtained by using saliva instead of blood [78], thus simplifying the whole procedure. Multiple sequential saliva samples can be easily obtained at home without the assistance of a health professional, avoiding invasive blood draws in a hospital setting. The DEX concentration should be determined at the same time as the cortisol to make sure that the test results are reliable. Lower or higher DEX concentrations might result in false-positive or false-negative results of nonsuppression.

### 6.4. The Decision Criterion

Carroll’s proposed criterion for the classification of the DST outcome (i.e., nonsuppression vs. suppression) is simply the cortisol threshold (cut-off) of 5 µg/dL. Other criteria, such as the difference or the quotient of basal (pre-DEX) and post-DEX cortisol, have been proposed, yielding only little improvement, if any. 

Recommendation: in addition to the post-DEX cortisol concentration, multiple other factors (e.g., DEX concentration, basal cortisol levels, cortisol metabolite, and other steroid levels [79]) and patient demographics (age, sex, etc.) could have an important impact on the outcome of the DST; these factors should be included in the building of a predictive model with optimal performance [80,81]. Another important aspect is the continuous nature of the cortisol concentration. Any binarization reduces the information content, so the numerical outcome should be used instead of a binary response (e.g., ≤5 vs. >5 µg/dL). Small deviations in post-DEX cortisol measurements close to the threshold value influence the outcome label, but it is evident that a reading of 4.99 (suppression) is not significantly different from 5.01 (nonsuppression). Even with the use of confidence intervals, the repetition of measurements close to the threshold or the use of averages does not eliminate this conceptual problem. Instead of the simple decision criterion proposed by Carroll, an advanced model that yields a case-wise probability of being a nonsuppressor would be advantageous [82,83].

### 6.5. Cortisol Secretion Pattern

The 24 h circadian secretion rhythm of cortisol is common knowledge [84], but the existence of an additional superimposed ultradian rhythm is less well known. Figures of the cortisol secretion patterns showing these two rhythms based on high-frequency cortisol measurements in healthy controls, depressed patients, and patients with hypercortisolism (Cushing’s disease/syndrome) can be found in the literature [85,86,87,88]. These cortisol profiles indicate that there is a large interindividual variance in the cortisol levels at any given time. Cortisol is secreted in pulses of a rather high amplitude. Linkowski et al. reported absolute cortisol pulse amplitudes of 6.8, 6.3, and 7.8 µg/dL for normal controls, and unipolar and bipolar depressed patients, respectively [89]. This is also true for the afternoon post-DEX period, when the most important cortisol measurements were made. This observation can have a huge impact on the outcome of the DST, as the timing of the pulses is variable and the difference between cortisol concentrations at the peak of a pulse and the baseline can be substantial. Mark Gold et al. conducted a DST study including 65 patients with “primary major depression” in which cortisol was quantified at six different times between 8 a.m. and 12 p.m. post-DEX. They found that when applying Carroll’s standard procedure (two measurements post-DEX at 4 and 11–12 p.m.), 31% of the patients were found to be nonsuppressors. When all six cortisol measurements were used (i.e., cortisol concentration at any time point > 5 µg/dL), the proportion of nonsuppressors rose to 44% [90,91]. Carroll et al. published an in-depth study of the pulsatile secretion of cortisol and ACTH in depressed patients using high-frequency blood sampling [92], in which depressed patients with and without hypercortisolism were clearly distinguished. In hypercortisolemic depression (i.e., severe depression with hypercortisolism), cortisol secretion is irregular and is uncoupled from ACTH secretion. 

Recommendation: using the average or maximum of multiple measurements during a period of several hours in the afternoon post-DEX can solve this problem.

### 6.6. Factors Influencing the Sensitivity of the DST

Carroll found a diagnostic sensitivity of the DST in melancholic patients of 67%, but most researchers reported significantly smaller sensitivities, probably due to the fact they used less stringent diagnostic criteria for the identification of melancholic patients such as the Research Diagnostic Criteria (RDC) and the DSM-III. Some researchers even understood MDD as a synonym of melancholia, leading to a dilution and substantial reduction in the proportion of real melancholia cases in their patient cohorts. In this sense, the introduction and promotion of the DSM-III in 1980 by the APA had a very negative impact on the performance and acceptance of the DST. The use of any etiology-agnostic diagnostic tool with nonvalidated entities has had a disastrous effect on research in psychiatry and psychopharmacology [93].

Various technical factors, such as the cortisol threshold being too high, incorrect timing, and insufficient sampling, impact the sensitivity of the DST. If the cortisol threshold is too high or the timing is wrong (e.g., in the morning) or only a single cortisol determination is taken, the sensitivity of the DST is lowered. The use of too little DEX gives rise to false-positive nonsuppression. Another very important factor causing artificially high proportions of nonsuppressors is a too-low post-DEX dexamethasone concentration. This has been observed by multiple researchers and even Carroll himself. However, this important topic was never factored into an improved DST. For a DST to be valid, an acceptable range of the dexamethasone concentration (post-DEX) at different times must be defined, and the dexamethasone and cortisol concentrations must be contemporaneously determined together [69].

Another interesting observation is the fact that even Carroll obtained a DST sensitivity of only 67%. Why were the remaining third not also nonsuppressors? Obviously, the functionality of their HPA axis was intact, and they did not have a pronounced hypercortisolism. This strongly indicates that the group of patients with melancholia was biologically not homogenous.

In his publication Neuroendocrine Probes as Biological Markers of Affective Disorders, Canadian psychiatrist Gregory Brown reviewed five endocrine systems: the hypothalamic–pituitary–adrenal axis, hypothalamic–pituitary–thyroid axis, growth hormone regulation, prolactin regulation, and pineal function [94]. Abnormalities in all these systems were found in depressed patients.

Multiple endocrine axes in the same cohort of depressed patients have been assessed in several studies, two of which are reviewed below:Extein et al. applied both the thyrotropin-releasing hormone (TRH) test and the DST (1 mg DEX, cortisol measurements at 8 a.m., noon, 4 p.m., and midnight post-DEX; threshold: ≥6 µg cortisol/dL) to a cohort of 50 inpatients with unipolar depression, using the RDC as the diagnostic tool. All patients were euthyroid and without evidence of endocrine disease. A total of 84% of the patients showed a dysfunction of the HPA or HPT axis, 34% of the HPT axis only, 20% of the HPA only, and 30% of both axes. Of the patients, 64% had a blunted TSH response to TRH, and 50% failed to suppress on the DST [90].Gordon Parker et al. assessed the function of three different endocrine axes in 40 inpatients meeting the DSM-III-R criteria for MDD with melancholia (19 with and 21 without psychosis). Of the patients, 80% showed disturbances in at least one hormonal axis, 40% in two axes, and 5% in all three axes. Growth hormone (GH) blunting was found in 62.5%, DST nonsuppression in 37.5%, and TSH blunting in 25.0% [95].

Obviously, the melancholia phenotype, as psychopathologically defined, consists of subgroups with different pathophysiologies. As stated earlier, various somatic causes can produce a clinical picture resembling melancholia. Among them are autoimmune and endocrine diseases, neoplasms (in particular carcinomas of the pancreas), and infections, in short, anything causing a dysfunction of the brain. In melancholic patients, multiple hormonal axes can be individually or simultaneously disturbed [90,95]. This emphasizes the importance of a biochemical identification of the various subgroups. The application of a single laboratory test, e.g., the DST, allows for the identification of a biologically more homogenous subgroup of melancholic patients. On the psychopathological level alone, this cannot be achieved. The use of a biomarker such as the DST provides an additional important element. In biomarker research, this is called phenotypic anchoring. Multiple biomarkers/tests can be used for the identification of different homogenous subgroups.

### 6.7. Factors Influencing the Specificity of the DST

Zimmerman and Coryell presented a review illustrating the results of 53 studies in which the 1 mg dexamethasone suppression test in normal controls [96] was applied. A mean rate of nonsuppression at 4 p.m. of 7.4% and 6.3% at 11 p.m. was found. However, in 11 (21%) of these studies, the reported rates of nonsuppression were higher than 10%. Factors such as recent weight loss, sleep deprivation, psychosocial stress, caffeine use, and possibly older age can cause nonsuppression in normal controls [96,97].

Even more disturbing is the observation that elevated rates of nonsuppression were found in psychiatric patients with diagnoses other than melancholia. Carroll claimed the DST to be a specific test for the diagnosis of melancholia, so patients with other nonorganic forms of depression, such as neurotic/reactive depression, should not show nonsuppression. However, substantially elevated rates of positive DSTs (i.e., nonsuppression) were found also in patients with personality disorders, schizophrenia, mania, anxiety disorders, post-traumatic stress disorder (PTSD), substance abuse (alcoholism), and dementia [23,55]. Obviously, a hyperactivity of the HPA axis can have many different causes, even purely psychological ones.

How should all these findings be taken into consideration, and what is the impact of these results on the applicability of the DST? To do justice to Carroll, it must be emphasized that he applied the DST to a very stringently defined patient population, and many patients with confounding medical problems were excluded. In addition, the test was never proposed as a screening test for a broad patient population with a wide range of psychiatric disorders. The DST should be applied only to a well-defined melancholic patient population with severe mood disorders (in the broad sense of Kraepelin’s manic-depressive insanity (MDI)). Additionally, the current clinical picture should be that of melancholia [37,38]. Other bordering conditions, such as mixed states, schizoaffective disorder, and atypical psychoses, could all belong in a certain sense to the MDI fold with a corresponding high proportion of nonsuppressors.

Although schizophrenia subtypes are no longer considered in DSM-5, schizophrenia is still recognized by many experts to be a very heterogeneous group of disorders. Some of the (old) subtypes, such as hebephrenia and schizophrenia simplex, are definitively distinct from others such as paranoid and catatonic schizophrenia. Catatonia has more recently been shown to be a disease in its own right. Therefore, a dissection of the schizophrenia pool should be seriously considered. A part of this “pool” should probably be relabeled as a psychotic mood disorder [98]. Therefore, elevated rates of nonsuppression in any of the disease groups mentioned above would not be an unexpected or a contradicting finding. We should not forget that symptomatic/secondary schizophrenias can have many different somatic causes, not only a hyperactive HPA axis [99,100].

## 7. Possible Applications of an Improved and Validated DST

The DST as a biomarker (BM) in psychiatry has multiple potential applications. As clinical pictures of mental disorders are unspecific and do not allow for the unequivocal identification of their etiology, any additional information strengthening the diagnosis is helpful. Originally developed for diagnostic purposes (diagnostic BM), it can also be applied to predict the response to somatic treatments such as ECT and medication with TCAs (predictive BM), as DST nonresponders have a higher probability of a successful treatment [101]. Patients with a still-positive DST (i.e., nonsuppression) after a remission have a much higher risk of relapse than patients with a normalized DST. Therefore, the DST can also be used for the confirmation of a clinical remission and prognosis of relapse [102,103,104].

Another important application of the DST is based on its link with suicide (prognostic BM). In a number of studies [105,106,107,108], depressed patients with a positive DST were more likely to have suicidal intentions, be hospitalized for suicide, and to complete suicide. Coryell and Schlesser [109], studying a group of depressed patients, highlighted that DST nonsuppression was associated with a 14-fold increased risk of eventual suicide. More recently, Ambrus [110] used the DST when investigating the relationship between leptin, HPA-axis activity, and anxiety in sixty-nine individuals with a recent suicide attempt.

The greatest value of the DST could be in the field of drug development. The use of symptom-based/etiology-agnostic diagnostic systems, such as the DSM and ICD, and the lack of robust biomarkers in psychiatry are the main obstacles to progress in the development of new psychotropic drugs. These factors are the cause of the heterogeneous patient populations used in drug development and the resulting meager results [111]. In addition to carefully characterized patients on the psychopathology level, any robust biomarker, such as an improved DST, would be very helpful in the generation of biologically more homogenous patient populations, thus enhancing the chances of success [112].

As indicated by the significant proportion of patients with treatment-resistant depression and the limited efficacy of modern antidepressants, a dysfunction of cerebral neurotransmitters cannot be the (only) cause of depression. There are many more potential organic causes of severe depressions, a dysfunctional HPA axis being one of them. Based on the assumption that a hyperactive HPA axis is at the core of the pathophysiology of melancholia, new avenues for its somatic treatment open up. Beverley Murphy from the University of Montreal treated depressed patients who were resistant to antidepressants with different antiglucocorticoids and obtained encouraging results [24,113]. There are now a substantial number of drugs available for the treatment of hypercortisolism [114,115,116]. They work on different targets of the HPA axis: (i) inhibition of cortisol biosynthesis in the adrenals (e.g., ketoconazole, metyrapone, osilodrostat, mitotane, and etomidate); (ii) pituitary (e.g., cabergoline and pasireotide); and (iii) blocking of the glucocorticoid receptor (mifepristone/RU486). These drugs could be used alone or in combination in cases of severe depression with a biochemically confirmed hypercortisolism and/or a positive DST. Some of the drugs mentioned above have been used in multiple studies for the treatment of depression with mixed results. Most of these studies are flawed, as the hormone status of the patients with regard to the functionality of the HPA axis was not determined, resulting in a biologically heterogenous patient population. Indeed, using a positive DST as an inclusion criterion for such trials with depressed patients would be very beneficial.

The very informative online tool BiGTeD (Biomarker-guided trial designs) covers all aspects of biomarker-enriched clinical trials and contains an exhaustive list of relevant references [117].

## 8. Conclusions and Outlook

The DST as proposed by Carroll et al. in 1981 was never fully developed, validated, or standardized. A plethora of investigations by other research groups applying the DST yielded contradictory findings, which resulted in it finally being rejected. In addition, many technical and political issues, diagnostic fuzziness, and the prevailing *zeitgeist* contributed to the fall from grace of this promising biomarker [1,6].

The increase in knowledge of endocrinology over the last 40 years with regard to the HPA axis and its disorders, as well as new developments in clinical chemistry and laboratory medicine, could contribute to generating a new version of the DST. In this regard, LC-MS/MS technology for the simultaneous quantification of multiple hormones, which might enhance the test performance [79], is particularly worthy of mention. This also allows for a noninvasive simultaneous quantification of multiple hormones in saliva [68,78]. After a careful improvement and validation an updated DST has the potential to become a useful tool in psychiatry’s armamentarium [118].

As discussed in this review, there are many experimental parameters and statistical issues which need to be considered in the development of an improved DST. Another important aspect is the careful psychopathological characterization of the patients to be tested. It must be emphasized that the current diagnostic concept of major depressive disorder (MDD) as found in the DSM-5-TR and ICD-11 should not be confused with melancholic depression [35,36]. The DST was and is intended to be used only in cases of melancholia (in the broad sense of Kraepelin’s manic-depressive insanity). We emphasize the fact that the DST should be applied only to patients with the psychopathological picture of a severe depression in a broad sense (including psychotic depression, bipolar disorder, and schizoaffective disorder). Patients with somatic/psychological conditions linked to hypercortisolism should not be tested.

An improved and fully validated DST has several possible applications in clinical psychiatry. In addition to its primary use as a diagnostic tool, it could also be useful for the assessment of suicidality, prediction of response to treatment, confirmation of clinical remission, and prognosis of relapse.

Maybe most importantly, the DST could become a crucial part in the generation of biologically homogeneous patient populations and thus considerably enhance the chances of psychotropic medications being successfully developed.

## Figures and Tables

**Table 1 jpm-13-00837-t001:** Sensitivities of the DST for the detection of MDD-ET (controls = healthy subjects; other = patients with psychiatric diagnoses other than MDD-ET).

			Nonsuppressors (%)		
Study	Dose (mg)	Sampling Time	Depressed	Controls	Other
Stokes 1984	1	8 a.m.	26	10	37
Coppen 1983	1	4 p.m.	81	11	35
Carroll 1980	1	4 p.m.	35	4	-
Carroll 1981	1 + 2	8 a.m., 4 p.m., 11 p.m.	-	-	4
Amsterdam 1982	1	4 p.m.	25	15	-
Schlesser 1980	1	8 a.m.	43	-	0
Brown 1979	2	8 a.m., 4 p.m., 11 p.m.	40	-	0
Holsboer 1980	2	4 p.m.	22	-	14
Average			39	10	15

**Table 2 jpm-13-00837-t002:** Percentages of DST nonsuppression in psychiatric inpatients with different diagnoses.

		DST Threshold (µg/dL)		
Diagnosis	≥4	≥5	≥10	≥15
Mixed bipolar	100	100	43	14
Psychotic depression	95	95	84	47
Melancholia	87	78	43	9
Organic affective syndrome	67	67	33	33
MDD (no melancholia)	60	48	21	6
Schizoaffective	43	43	29	0
Mania	33	33	33	33
“Depressive symptoms”	19	14	3	3

**Table 3 jpm-13-00837-t003:** Percentages of DST nonsuppressors in various psychiatric populations.

Diagnosis	Nonsuppressors (%)
Mixed bipolar	78
Psychotic depression	67
Melancholia	50
Dementia	41
Mania	41
Bipolar	38
Psychosis	34
Minor depression	23
Schizophrenia	13
Anxiety	8
Normal controls	7

## Data Availability

Not applicable.

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
