# Peer review of "Cortisol and the Dexamethasone Suppression Test as a Biomarker for Melancholic Depression: A Narrative Review"

_jpm, 2023, doi:10.3390/jpm13050837_

Round 1
Reviewer 1 Report
Cortisol and the Dexamethasone Suppression Test as a Biomarker for Melancholic Depression. A Narrative Review
In general, the manuscript is clear and understandable. After revision, I only observed minimal errors that should be checked.
In page 2, line 95
In certain subclinical forms of endogenous CS [17-18], psychiatric symptoms are thei first and only manifestation (19).
What do you want to write: the or their. Checked it, please.
In page 5, line 255
Clinical uses: assessment of treatment response to ECT or pharmacotherapy with TCAs,
Please, explain what are ECT and TCAs, describe acronym.
In page 8, line 344
The proportion of non-responders correlated with the severity of the depression, as measured by HAMD.
Please, explain what is HAMD, describe acronym.
In page 10, line 438
and patient demographics (age, gender, etc.)) could have an important impact on the outcome of the DST
Please, erase the second parentheses sign.
In page 10, line 467
(i.e.; cortisol conc. at any time point > 5μg/dl)
Please, complete the word conc.
In page 11, line 479
the RDC and the DSM-III.
Please, explain what is RDC, describe acronym.
Cortisol and the Dexamethasone Suppression Test as a Biomarker for Melancholic Depression. A Narrative Review
In general, the manuscript is clear and understandable. After revision, I only observed minimal errors that should be checked.
In page 2, line 95
In certain subclinical forms of endogenous CS [17-18], psychiatric symptoms are thei first and only manifestation (19).
What do you want to write: the or their. Checked it, please.
In page 5, line 255
Clinical uses: assessment of treatment response to ECT or pharmacotherapy with TCAs,
Please, explain what are ECT and TCAs, describe acronym.
In page 8, line 344
The proportion of non-responders correlated with the severity of the depression, as measured by HAMD.
Please, explain what is HAMD, describe acronym.
In page 10, line 438
and patient demographics (age, gender, etc.)) could have an important impact on the outcome of the DST
Please, erase the second parentheses sign.
In page 10, line 467
(i.e.; cortisol conc. at any time point > 5μg/dl)
Please, complete the word conc.
In page 11, line 479
the RDC and the DSM-III.
Please, explain what is RDC, describe acronym.
Author Response
Answers to Reviewer 1.
In page 2, line 95
In certain subclinical forms of endogenous CS [17-18], psychiatric symptoms are thei first and only manifestation (19).
What do you want to write: the or their. Checked it, please.
Answer: psychiatric symptoms are their first and only manifestation.
Done
In page 5, line 255
Clinical uses: assessment of treatment response to ECT or pharmacotherapy with TCAs,
Please, explain what are ECT and TCAs, describe acronym.
Answer: Clinical uses: assessment of treatment response to electroconvulsive therapy (ECT) or pharmacotherapy with tricyclic antidepressants (TCAs)
Done
In page 8, line 344
The proportion of non-responders correlated with the severity of the depression, as measured by HAMD.
Please, explain what is HAMD, describe acronym.
Answer: The proportion of non-responders correlated with the severity of the depression, as measured by the Hamilton Depression Scale (HAMD).
Done
In page 10, line 438
and patient demographics (age, gender, etc.)) could have an important impact on the outcome of the DS
Please, erase the second parentheses sign.
Done
In page 10, line 467
(i.e.; cortisol conc. at any time point > 5μg/dl)
Please, complete the word conc.
Answer: Cortisol concentration
Done
In page 11, line 479
the RDC and the DSM-III.
Please, explain what is RDC, describe acronym.
Answer: used less stringent diagnostic criteria for the identification of melancholic patients such as the Research Diagnostic Criteria (RDC) and the DSM-III.
Done
Reviewer 2 Report
This review renews attention on the DST, several decades after it was examined in multiple studies as a possible biomarker for melancholia. The DST has considerable promise for clinical and research use, and this paper can help guide the field toward refreshing this potentially valuable tool.
I have some comments and questions:
Line 122: Define CD.
150-174: It's good that you describe melancholia because this term may be new to people outside the field. You might move this section earlier in the paper, before you start mentioning melancholia and DST. You might also add some more information on melancholia. Is there a more succinct description of melancholia than the presence of the 6 features you list in lines 170-174? I saw one paper describe the symptoms as unremitting gloom, psychomotor disturbance, and anhedonia, (but "psychomoter disturbance" is vague; another paper described it as retardation and agitation, which is a little clearer, but still rather abstract). What proportion of depression falls within the melancholia type? Since you mention definite family history, is melancholia more heritable than other forms of depression? What is its heritability (just briefly)? Is melancholia less likely to be preceded by environmental factors (loss or grief) than other forms of depression?
154: I'd never heard of the term "alienists" and had to look it up. For readers who know little of the history of psychiatry, like me, you might very briefly introduce the term.
164: Later you mention the CORE measure several times, so it's worth taking space in your paper to at least briefly summarize it.
170-174: I assume that all six of these (and not only one) are required for a diagnosis, but you may clarify this.
210: Were these pre-DEX measurements done on all the subjects? Why not mention this blood sampling in section 4.1? Would it be useful to compare pre-DEX and post-DEX measurements on the same individual, rather than just against a concentration of 5 μg/dl?
223-224: Can you elaborate a little on why and how you make this statement? What would be the difference between an episode-related biomarker of melancholia and a trait marker of melancholia?
225: Define ECT and TCA.
245-246: Perhaps you discussed this later in the paper, but could one solution to this problem be to approach it differently: whether the DST is actually useful rather than whether it can classify people according to some somewhat arbitrary definition? In other words, bypass a clinical definition of endogenous depression. Simply measure the characteristics of people based on DST results, and then examine whether the results can be directly useful for treatment or research.
257: The result of the DST always seems to be interpreted categorically: either suppression or non-suppression. But is there additional useful information to extract from the DST by considering the results quantitatively: the level of cortisol (or the change in the level of cortisol)? Maybe some patients have partial suppression, corresponding with mild melancholia. Forced lumping of such patients with either the normal or suppressed group would muddy the results. After I read further, to section 6.4, I saw that you had addressed this. Maybe you can mention this earlier, or refer to section 6.4.
263-264: I hadn't heard of the Research Diagnostic Criteria. Looking it up, I see that it preceded the DSM-III. It's good to know history, but not all readers will have studied it. If you think it's important that readers know aspects of psychiatry history, you can help inform them by briefly describing the Research Diagnostic Criteria and giving a citation.
Table 2: I'd never heard of Organic syndrome. Looking it up, I see that it seems to be either dementia, delirium, or amnesia, but you may briefly define it in the paper for readers who don't know it.
Tables 2 and 3: It's interesting that mixed bipolar and psychotic depression gave even more pronounced DST results than melancholia. You might comment more on this.
472-474: Could this repeated sampling be done using saliva, as discussed in Section 6.3? If so, you may mention that this would make it easier to collect samples at multiple times.
Very good: only a few corrections needed.
Author Response
Answers to Reviewer 2.
Line 122: Define CD.
Answer: Done on line 28: In healthy subjects, the release of CRH and ACTH is regulated by cortisol via a negative feedback mechanism, while in patients with Cushing’s disease (CD) and other subtypes of CS this feedback is impaired
Lines150-174: It's good that you describe melancholia because this term may be new to people outside the field. You might move this section earlier in the paper, before you start mentioning melancholia and DST. You might also add some more information on melancholia. Is there a more succinct description of melancholia than the presence of the 6 features you list in lines 170-174? I saw one paper describe the symptoms as unremitting gloom, psychomotor disturbance, and anhedonia, (but "psychomoter disturbance" is vague; another paper described it as retardation and agitation, which is a little clearer, but still rather abstract). What proportion of depression falls within the melancholia type? Since you mention definite family history, is melancholia more heritable than other forms of depression? What is its heritability (just briefly)? Is melancholia less likely to be preceded by environmental factors (loss or grief) than other forms of depression?
Answer:
Thank you for these valuable suggestions. A good understanding of the concept of melancholia is key to the whole issue of the DST. We have now introduced this concept in the introduction (see lines 54-58 below) and have added more details in section 3 (see lines 164-168 below).
Here are short answers to your other questions:
In contrast to exogenous (neurotic/reactive) depression, endogenous depression (i.e., melancholia) is considered to be a biological disease. The familial heritability of melancholia was estimated as 0.33 (Lamers et al. Familial aggregation and heritability of the melancholic and atypical subtypes of depression, J Affect Disord 2016;204:241-6).
Yes, melancholic episodes are less likely to be precipitated by external factors. They can, and do, often come “out of the blue”.
Melancholia has a prevalence in the low single-digit percent range in the general population. The prevalence of MDD is much higher. A rough estimate of the proportion of melancholia in a cohort of depressed patients is 5%.
Lines 54-58:
It is important to distinguish melancholic depression (melancholia, endogenous depression) from exogenous forms of depression (i.e., neurotic, reactive). Melancholia is considered to be a biological disease characterized by a recurrent course, familial aggregation, and a pronounced component of psychomotor disturbances (i.e., retardation and/or agitation of mental and physical activities).
Lines 164-168:
In contrast to ordinary depression, which is mainly characterized phenomenologically by different mental symptoms, melancholia has a strong somatic component comprising a recurrent course, familial aggregation and pronounced psychomotor disturbances. A melancholic episode often occurs without any external (psychosocial) triggers and is inert to psychotherapeutic interventions.
Line 154: I'd never heard of the term "alienists" and had to look it up. For readers who know little of the history of psychiatry, like me, you might very briefly introduce the term.
Answer: Melancholia (also called endogenous, endogenomorphic or vital depression) has been described as a clinical entity for millennia and was widely accepted by the physicians working in lunatic asylums of the past.
(Modified in the text at Line 160)
Line 164: Later you mention the CORE measure several times, so it's worth taking space in your paper to at least briefly summarize it.
Answer: We added a short description of the CORE measure (see lines 175-188 below)
Line 175
Currently the most reliable diagnostic tool for melancholia is probably the CORE measure [36-37] of psychomotor disturbance, which is based on 18 signs (not symptoms) assessed by an experienced clinician. These signs belong to the subscales of agitation (facial agitation, motor agitation, facial apprehension, stereotyped movement, verbal stereotypy), retardation (slowed speed of movement, slowing of speech, delay in motor activity, bodily immobility, delay in verbal responses, facial immobility, postural slumping) and non-interactiveness (non-reactivity, inattentiveness, poverty of associations, shortened verbal responses, impaired spontaneity of talk). A value is attributed to each sign (0 if absent, 1 if present). These values are then added to get the total CORE score (range: 0 to 18). A score of 8 or more is needed for the diagnosis of melancholia. There is also a more refined CORE measure, graded according to the severity of the signs (absent: 0, present: 1 to 3) available (range: 0 to 54). The reliability and validity of the CORE measure has been assessed in many international studies [37,59].
Lines 170-174: I assume that all six of these (and not only one) are required for a diagnosis, but you may clarify this.
Answer:
Regarding the adopted clinical diagnosis of melancholia [39], Carroll and colleagues wrote: ”The clinical features employed in the CSU for the diagnosis of endogenous depression are listed in Table 1. Rather than adopt a fixed number of features as a diagnostic criterion we require that the illness should be perceived by the psychiatrist as consistent with the syndrome ordinarily recognized as endogenous depression or melancholia.”
As no fixed number of criteria must be met, the diagnosis depends on the clinical experience of the physician and is somewhat subjective.
Therefore, we prefer and recommend the more objective and quantitative CORE measure for the diagnosis of melancholia/endogenous depression.
Line 210: Were these pre-DEX measurements done on all the subjects? Why not mention this blood sampling in section 4.1? Would it be useful to compare pre-DEX and post-DEX measurements on the same individual, rather than just against a concentration of 5 μg/dl?
Answer:
Yes, the pre-DEX cortisol measurements were done on all subjects. However, Carroll and colleagues did not find these measurements to be helpful with regard to the performance of the DST (see section 4.2., line 213). The pre-DEX (=basal) cortisol levels are not part of the original DST as proposed by Carroll and therefore are not mentioned in section 4.1.
Different researchers have used both the pre- and post-DEX cortisol levels of individual patients (as differences or quotients). This is described in section 6.4.
We are not in favor of using simply the 5 μg/dl cortisol threshold as a decision criterion of an improved DST. We recommend considering also the pre-DEX cortisol level, among other factors, in the development of an improved version of the DST (see section 6.4.).
Lines 223-224: Can you elaborate a little on why and how you make this statement? What would be the difference between an episode-related biomarker of melancholia and a trait marker of melancholia?
Answer: This is now explained in more detail in the text (section 4.3., lines 249-253, see below).
Line 249:
The DST is a specific episode-related biological marker of melancholia (i.e., it is a state-dependent biomarker, not a trait marker of melancholia per se). A diagnostic trait biomarker would identify the disease at all times, a state biomarker only when the disease is active. This distinction is very important in the case of episodic diseases like melancholia. An additional benefit of the DST as a state biomarker is its usefulness to monitor the success of treatment (i.e., normal suppression in remission).
Line 225: Define ECT and TCA.
Answer: Now in the text (section 4.3., lines 254-255 below).
Clinical uses: assessment of treatment response to electroconvulsive therapy (ECT) or pharmacotherapy with tricyclic antidepressants (TCAs)
Lines 245-246: Perhaps you discussed this later in the paper, but could one solution to this problem be to approach it differently: whether the DST is actually useful rather than whether it can classify people according to some somewhat arbitrary definition? In other words, bypass a clinical definition of endogenous depression. Simply measure the characteristics of people based on DST results, and then examine whether the results can be directly useful for treatment or research.
Answer:
This question is very relevant. The outcome of the DST is rather unspecific. Many other medical conditions also yield positive results (i.e., non-suppression; compare section 6.7.). Therefore, we recommend that the DST be used only with patients with the psychopathological picture of a severe depression in a broad sense (including psychotic depression, bipolar disorder, and schizoaffective disorder).
This is recommendation is now described in the “Conclusions and outlook” section (lines 670-673 below).
The DST was and is intended to be used in cases of melancholia (in the broad sense of Kraepelin’s Manic-Depressive Insanity). We would recommend that the DST be used only with patients having the psycho-pathological picture of a severe depression in a broad sense (including psychotic de-pression, bipolar disorder, and schizoaffective disorder). Patients with somatic/psychological conditions linked to hypercortisolism should not be tested.
Line 257: The result of the DST always seems to be interpreted categorically: either suppression or non-suppression. But is there additional useful information to extract from the DST by considering the results quantitatively: the level of cortisol (or the change in the level of cortisol)? Maybe some patients have partial suppression, corresponding with mild melancholia. Forced lumping of such patients with either the normal or suppressed group would muddy the results. After I read further, to section 6.4, I saw that you had addressed this. Maybe you can mention this earlier, or refer to section 6.4.
Answer:
You are right. The use of a categorical outcome is a shortcoming of the DST.
We feel that the introduction and discussion of a continuous outcome in section 6.4. is appropriate.
Lines 263-264: I hadn't heard of the Research Diagnostic Criteria. Looking it up, I see that it preceded the DSM-III. It's good to know history, but not all readers will have studied it. If you think it's important that readers know aspects of psychiatry history, you can help inform them by briefly describing the Research Diagnostic Criteria and giving a citation.
Answer:
Yes, the RDC was the forerunner of the DSM-III. The RDC criteria are explained in reference [43], which is cited in lines 200 and 207. We do not believe that a further detailed description of the RDC criteria is needed.
Line 199:
Other researchers mostly used only symptom-based tools such as the Research Diagnostic Criteria (RDC) or the DSM-III, though some preferred the Newcastle scale [40-43].
Table 2: I'd never heard of Organic syndrome. Looking it up, I see that it seems to be either dementia, delirium, or amnesia, but you may briefly define it in the paper for readers who don't know it.
Answer:
The organic syndrome, or more precisely the organic affective syndrome, is fully explained in reference 57. It refers to an organic etiology of depression (e.g., hypothyroidism, vitamin B12 deficiency).
For an explanation of the different diagnostic labels listed in Table 2 the reader is now referred to reference 57. See lines 303-304.
“Organic syndrome” has been changed to “organic affective syndrome” (Table 2, line 307).
Line 303:
Table 2 below shows the percentages of non-suppression for a series of psychiatric disorders and multiple threshold values of the DST (adapted from Evans and Nemeroff [57]. The definition of the different diagnoses in the table can be found in this article).
Tables 2 and 3: It's interesting that mixed bipolar and psychotic depression gave even more pronounced DST results than melancholia. You might comment more on this.
Answer:
Indeed, this is an interesting finding. Obviously, all these conditions share the same pathophysiology. This indicates that melancholia, mixed bipolar and psychotic depression are not independent (categorical) diseases, but belong to a continuum of increasing severity. This has been added to the text (lines 311-314).
Line 311:
Patients with mixed bipolar, psychotic depression and melancholia show the highest rates of non-suppressors [58]. This indicates that these conditions are not independent (categorical) diseases, but are part of a continuum.
Lines 472-474: Could this repeated sampling be done using saliva, as discussed in Section 6.3? If so, you may mention that this would make it easier to collect samples at multiple times.
Answer:
Yes. Now mentioned more explicitly in section 6.3 (line 459 - 461).
Line 459:
Concentrations of free cortisol (and dexamethasone) could be obtained directly by using saliva instead of blood [78], thus simplifying the whole procedure. Multiple sequential saliva samples can easily be obtained at home without the assistance of a health professional, avoiding invasive blood draws in a hospital setting.